# Esophagectomy Versus Active Surveillance After Complete Response in Locally Advanced Esophageal Cancer: Retrospective Analysis

**DOI:** 10.3390/cancers17243926

**Published:** 2025-12-08

**Authors:** Efrat Gur, Meroslav Lutsyk, Tomer Meirson, Noor Abu Hjool, Dror Limon, Yosef Landman, Oded Icht, Baruch Brenner, Yulia Kundel

**Affiliations:** 1Davidoff Cancer Center, Rabin Medical Center–Beilinson Hospital, Petach Tikva 4941492, Israel; tomermei@clalit.org.il (T.M.); drorl@clalit.org.il (D.L.); yosefla@clalit.org.il (Y.L.); oded.icht@clalit.org.il (O.I.); brennerb@clalit.org.il (B.B.); 2Department of Oncology, Emek Medical Center, Afula 1834111, Israel; miroslavlut@clalit.org.il; 3Gray Faculty of Medical & Health Sciences, Tel Aviv University, Tel Aviv 6997801, Israel; norabu1@clalit.org.il

**Keywords:** esophageal cancer, active surveillance, esophagectomy, age-specific management

## Abstract

Esophageal cancer is an aggressive malignancy associated with poor survival outcomes. The current standard for locally advanced disease combines chemotherapy and radiotherapy followed by surgery. In a subset of patients, this combined approach can achieve complete tumor regression on imaging and biopsies. For such patients, the necessity of proceeding with surgery remains uncertain, given the substantial risks and postoperative morbidity associated with esophagectomy. In our study, we compared patients who underwent surgery after achieving a complete response with those who were managed through structured surveillance. Our findings demonstrate that younger patients derived a significant survival advantage from surgery, whereas in older patients, surveillance was safe and sometimes associated with better outcomes. These results highlight the need for age-adapted treatment strategies in esophageal cancer, aiming to maximize survival benefits in younger patients while minimizing unnecessary risks in the elderly.

## 1. Introduction

Esophageal cancer (EC) is a highly lethal malignancy characterized by both local invasion and distant dissemination [1,2]. Globally, it ranks as the eighth most common cancer and the sixth leading cause of cancer-related mortality [3,4,5]. The standard treatment paradigm for locally advanced disease involves either neoadjuvant chemoradiotherapy (nCT) or perioperative chemotherapy, both followed by esophagectomy [6,7,8,9], resulting in 5-year overall survival (OS) rates of 36–60% [10,11,12,13,14,15]. Survival outcomes vary by histologic subtype, with patients with squamous cell carcinoma (SCC) achieving higher pathological complete response (pCR) rates and superior long-term survival—with 5-year OS rates approaching 50–60% following nCRT plus surgery, whereas adenocarcinoma demonstrates lower response rates and 5-year OS rates of roughly 35–45%, even with modern perioperative regimens such as FLOT [10,11,12,16,17,18]. The landmark CROSS trial established the superiority of nCT plus surgery over surgery alone, demonstrating improved 5-year OS (47% vs. 34%) and notable rates of pCR, particularly in SCC (49%) [10].

In recent years, several studies have questioned the necessity of esophagectomy in patients achieving a clinical complete response (cCR) following chemoradiotherapy, particularly among those with SCC [17,18]. A meta-analysis comparing chemoradiotherapy followed by active surveillance with standard esophagectomy demonstrated comparable OS [19]. Similarly, a recent prospective study reported no significant differences in survival or locoregional failure rates between surgery and active surveillance in patients achieving cCR after nCRT [20]. Consistent with these findings, a multicenter propensity-matched study observed no significant differences in 3-year OS and DFS between patients managed with active surveillance and those undergoing immediate surgery [21,22].

The SANO trial, which compared active surveillance with standard esophagectomy in patients achieving a cCR after nCRT, demonstrated non-inferior survival between the two groups [23]. Retrospective studies of clinically complete responders managed with either active surveillance or immediate surgery have similarly shown no association with higher rates of distant dissemination or more severe adverse outcomes when surgery was deferred [22,24,25]. Building on these findings, we aimed to conduct a broader real-world cohort analysis to complement the controlled setting of prospective trials. Accordingly, we performed a retrospective single-institution study comparing survival outcomes in patients with cCR following nCRT who either underwent surgery or were managed without surgical intervention.

## 2. Materials and Methods

Data were retrospectively collected for patients treated with curative intent for locally advanced esophageal cancer at the Davidoff Cancer Center, Rabin Medical Center, between January 2013 and December 2023. The inclusion criteria were as follows: (1) age 18 years or older, (2) thoracic squamous cell carcinoma or adenocarcinoma, (3) locally advanced stages, defined as cT2-4aN+M0 according to the 8th edition of the American Joint Committee on Cancer (AJCC) TNM classification, (4) cCR, defined by chest computed tomography (CT), upper gastrointestinal (GI) endoscopy with biopsies, and positron emission tomography (PET) scan results after nCRT, (5) who were treated with neoadjuvant chemoradiotherapy (nCRT) followed either by esophagectomy or by active surveillance. This study was conducted at a high-volume tertiary medical center specializing in complex surgical procedures, including esophagectomy. The gastroenterologists involved in patient evaluation and follow-up are among the most experienced in the country, are recognized national leaders in advanced endoscopy, and utilize state-of-the-art endoscopic technologies. Approximately three expert gastroenterologists with dedicated expertise in this field were involved in the management of these patients.

Exclusion criteria comprised cervical esophageal cancer, missing medical records, and metastatic disease at diagnosis.

The nCRT protocol included two chemotherapy regimens. The first consisted of two intravenous cycles of cisplatin (100 mg/m^2^) on days 1 and 29, combined with continuous 24-h infusion of 5-FU (800–1000 mg/m^2^) on days 1–4 and 29–32 of a 35-day cycle. The second regimen followed the CROSS protocol, consisting of weekly carboplatin (AUC 2 mg/mL/min) and paclitaxel (50 mg/m^2^) administered on days 1, 8, 15, 22, and 29. In the nCRT setting, radiotherapy was delivered concurrently to a total dose of 41.4 Gy in 23 fractions of 1.8 Gy each, with five fractions per week as per the CROSS protocol. For definitive chemoradiotherapy, the total dose was 50.4 Gy.

Restaging was performed four weeks after completion of nCRT with PET-CT imaging. Criteria for cCR included: (1) radiographic tumor resolution on CT, (2) a reduction in maximum standardized uptake value (SUVmax) >35% on PET-CT, and (3) absence of residual tumor on endoscopy with negative biopsy. Treatment strategies were subsequently discussed in a multidisciplinary tumor board comprising medical oncologists, radiation oncologists, surgeons, and radiologists. Patients deemed medically unfit for surgery due to significant comorbidities were managed with definitive chemoradiotherapy (50.4 Gy concurrent with chemotherapy). Additional patients who declined surgery after achieving cCR were also managed with active surveillance.

Patients in the active surveillance cohort underwent PET-CT and endoscopy every three months. Endoscopic findings suspicious for recurrence were biopsied for histological confirmation. Patients in the surgical cohort underwent esophagectomy with curative intent, using either open or minimally invasive techniques. Postoperative surveillance included PET-CT every three months during the first two years, every six months during the subsequent three years, and annually thereafter, combined with routine endoscopy. Biopsies were obtained for suspicious endoscopic findings. The follow-up protocol was consistent across both surveillance and surgical groups.

Clinical and histopathological data were retrieved from institutional databases, including demographic characteristics, the date of diagnosis, follow-up details, and the most recent clinic visit. Tumor histology, anatomical location, and TNM staging were recorded based on endoscopic ultrasound (EUS) and PET-CT assessments. Data on initiation of chemoradiotherapy, chemotherapy regimen, radiation dose and fields, surgical details, and pathological reports were collected. Follow-up information included PET-CT findings, recurrence dates, survival status, and date of last clinic visit.

### Statistical Analysis

Baseline characteristics and postoperative outcomes were summarized as proportions (percentages) for categorical variables and as medians with interquartile ranges (IQR) for continuous variables. Continuous variables were compared using Student’s *t*-test, while categorical variables were analyzed using the chi-square (χ^2^) test. Fisher’s exact test was employed when comparing two categorical variables or when expected frequencies were low.

Median follow-up duration was estimated using the reverse Kaplan–Meier method, accounting only for patients who remained alive. Survival outcomes were analyzed using Kaplan–Meier curves for both the active surveillance and immediate surgery groups. Comparisons between groups were performed using the log-rank test, and results were expressed as hazard ratios (HRs) with 95% confidence intervals (CI). All statistical tests were two-sided, and a *p*-value < 0.05 was considered statistically significant. Analyses were performed using SAS software, version 4.9 (SAS Institute Inc., Cary, NC, USA).

## 3. Results

### Baseline Characteristics

During the study period, 252 patients with locally advanced esophageal cancer were treated with nCRT. Among them, 118 achieved a cCR, including 64 patients (54%) with SCC and 54 patients (45%) with adenocarcinoma (AC). Of these 118 patients, 70 underwent esophagectomy, while 48 continued on active surveillance without surgery, primarily due to medical comorbidities or refusal of surgical intervention. Among the surgical cohort, 33 patients (47%) achieved a pathological complete response (pCR).

Among the 66 patients who underwent esophagectomy, postoperative complications were observed in several domains. Pulmonary complications occurred in 10 patients (15.1%), with pneumonia accounting for 15% of all reported complications. Respiratory failure was documented in 2 patients (3.0%). Anastomotic leakage was identified in 20% of surgical patients (13 of 66 patients). In addition, deep vein thrombosis (DVT) occurred in 3 patients (4.5%). These findings reflect the substantial morbidity associated with esophagectomy in this cohort.

Overall, 63 patients (53%) were male and 55 (46%) female (Table 1). Patients in the surveillance group were significantly older compared with those undergoing surgery (68% vs. 37.1%, *p* < 0.001). SCC was more frequent in the surveillance cohort (81.2% vs. 35.7%, *p* < 0.001). Advanced tumors (T3–T4) were common in both groups but more prevalent in the surgical group (90% vs. 75%, *p* = 0.04). The majority of patients in both groups had N1 nodal disease (66.6% surveillance vs. 68.5% surgery). A higher proportion of patients in the surveillance group received a definitive radiation dose, though this difference was not statistically significant (64.5% vs. 58.5%, *p* = 0.56).

With regard to neoadjuvant protocols, the CROSS regimen was administered in 47.9% of the surveillance group and 42.8% of the surgical group. Cisplatin plus 5-FU was administered in 41.6% of surveillance patients compared with 14.2% of surgical patients.

The 5-year OS rate was 48% in the surveillance group and 49% in the surgical group. 5-year DFS rates were 36% and 43%, respectively. Median OS was 2.57 years in the surveillance group and 2.63 years in the surgical group, and median DFS was 2.2 years in both groups.

During follow-up, 32 patients (68%) in the surveillance group died, 11 (23%) without evidence of disease (NED). In the surgical cohort, 37 patients (53%) died, including 5 (7%) NED.

No significant differences were observed in DFS (HR = 0.88, 95% CI 0.55–1.41, *p* = 0.25) or OS (HR = 0.75, 95% CI 0.47–1.26, *p* = 0.27) between the surgery and surveillance groups (Figure 1). Subgroup analyses by histology (AC vs. SCC) revealed no significant differences in either OS (*p* = 0.76 for AC, *p* = 0.35 for SCC) or DFS (*p* = 0.23 and *p* = 0.52, respectively) (Figure 2 and Figure 3).

When stratified by age, patients aged ≤70 years demonstrated a significant benefit from surgery compared with surveillance, with both improved disease-free survival DFS (HR = 0.44, 95% CI 0.20–0.90, *p* = 0.02) and OS (HR = 0.59, *p* = 0.032) (Figure 4).

In contrast, among patients >70 years, no significant differences in OS (HR = 1.4, 95% CI 0.7–2.6, *p* = 0.25) or DFS (HR = 1.44, 95% CI 0.7–2.6, *p* = 0.20) were observed; indeed, a trend toward inferior outcomes with surgery was noted in this older cohort (Figure 5).

In multivariable analysis, older age (*p* = 0.005) and female sex (*p* = 0.007) emerged as independent predictors of OS. No additional predictors were identified.

Recurrence rates were similar between groups, although local recurrences were more common in the surveillance cohort (12.5% vs. 8.5%). Overall recurrence rates were higher in the surveillance group (57.2% vs. 42.9%), though these differences did not reach statistical significance (*p* = 0.26).

## 4. Discussion

Neoadjuvant chemoradiotherapy (nCRT) has become the standard of care for treatment of locally advanced esophageal cancer, with landmark trials such as CROSS and NEOCRTEC5010 demonstrating improved overall and disease-free survival, as well as better local control, when nCRT is followed by surgery compared to surgery alone [10,26,27,28]. However, recent evidence has challenged the necessity of routine esophagectomy in all patients achieving a clinical complete response (cCR) after nCRT. Notably, retrospective studies and RCTs including the FFCD 9102 trial and analyses by van der Wilk et al. have shown no significant difference in OS between active surveillance and immediate surgery in selected patients with cCR, suggesting that non-operative management may be a viable alternative in appropriately selected individuals [19,29]. Furthermore, a meta-analysis evaluating the role of surgery in this setting indicated that nCRT alone was associated with superior survival outcomes compared with nCRT plus surgery, without significant differences in DFS [20].

This study aimed to assess whether active surveillance could be a safe alternative to surgery in patients with locally advanced esophageal cancer who achieved a cCR following chemoradiotherapy, without compromising survival outcomes. Our findings suggest that in patients with locally advanced esophageal cancer who achieve a cCR after chemoradiotherapy active surveillance may be a viable alternative to immediate surgery without compromising OS. Notably, in patients aged 70 years or younger, surgery was associated with a statistically significant improvement in DFS and OS compared with active surveillance.

The observation that the survival benefit associated with surgery in younger patients is linked with a corresponding improvement in OS, supports the consideration of surgery in younger patients. These findings underscore the importance of careful patient selection and clinical judgment when determining eligibility for non-operative management. Among patients older than 70 years, no differences in OS were observed between the surveillance and surgery groups. The benefit of surgery observed in younger patients may reflect selection bias, if younger individuals assigned to surveillance were, in fact, less fit for surgery—and their reduced fitness, rather than the lack of surgery, contributed to worse outcomes. Therefore, the observed effect may be attributable to confounding by indication rather than to the surgical intervention itself.

Our findings, showing no survival benefit from surgery in patients older than 70 years, are consistent with previous reports indicating that patients older than 75 years with SCC who achieved cCR did not derive additional benefit from surgery following chemoradiotherapy, whereas those with adenocarcinoma did demonstrate improved survival with surgical resection [30]. However, contrasting evidence from other studies suggests that trimodality therapy—combining nCRT with surgery—can significantly improve both OS and PFS compared with other treatment approaches, irrespective of patient age [31]. Therefore, this remains a controversial topic with no definitive conclusion. The key implication is that in an older patient population, a morbid surgical procedure can be avoided without compromising survival, ultimately preserving their quality of life. While several studies have compared nCRT followed by active surveillance vs. surgery, there is a lack of research specifically focusing on older patients. We believe that the findings of this study have significant implications for the management of this patient population, particularly for elderly patients, in whom surgery may potentially be avoided.

Among patients in the surgery group, a pathological complete response (pCR) was observed in 47%, indicating that current restaging modalities—such as CT, PET-CT, and endoscopic biopsy—underestimated residual disease in nearly half of cases. This discordance between cCR and pCR has been widely reported, with literature citing pCR rates among cCR patients ranging from 24% to 73%, likely due to heterogeneity in restaging approaches [12,17,18]. These findings highlight the need for standardized, rigorous restaging algorithms to more reliably identify true responders and guide treatment decisions. Recent evidence further suggests that even among patients achieving pCR, minimal residual disease may still be present when evaluated with advanced molecular or imaging-based techniques. A recent study demonstrated that circulating tumor DNA (ctDNA)-based monitoring can detect molecular residual disease in patients classified as pCR, identifying individuals at higher risk for recurrence despite histopathologic clearance. Such tools may complement conventional restaging and ultimately improve selection of candidates for active surveillance [32].

Local recurrence rates were higher in the active surveillance group compared with the surgery group, although this difference did not reach statistical significance. These findings are consistent with results from a systematic review and meta-analysis demonstrating no significant difference in distant metastatic recurrence between the two approaches, provided that cCR was confirmed and rigorous follow-up protocols were implemented [19]. Notably, the 5-year locoregional recurrence rate during active surveillance was approximately 40%, with 95% of patients undergoing successful R0 resection upon delayed surgery [19]. According to current literature, between 20 and 40% of patients who achieve a pCR following nCRT and surgery will experience disease recurrence, irrespective of histological subtype.

Current evidence suggests that while active surveillance may result in higher rates of locoregional recurrence compared to immediate surgery after chemoradiotherapy, the rates of distant metastatic relapse appear similar between groups [33]. Data from trials such as SANO and recent meta-analyses indicate that patients managed with active surveillance following a cCR experience local regrowth in approximately 40–50% of cases; however, most of these recurrences are amenable to timely salvage surgery, preserving overall survival outcomes [34,35]. In contrast, the rate of distant metastatic failure does not appear to differ significantly between active surveillance and immediate surgery cohorts, suggesting that surgery primarily affects local disease control rather than systemic progression. This suggests that there are three distinct patient groups that may benefit from active surveillance: (1) patients with aggressive tumors who ultimately experience distant metastatic failure and have therefore avoided unnecessary morbidity from surgery; (2) patients effectively cured by nCRT, eliminating the need for surgery entirely; and (3) patients who experience locoregional recurrence but benefit from an extended surgery-free interval compared to those undergoing immediate surgery. These findings underscore the importance of rigorous follow-up protocols in active surveillance strategies, allowing early detection and intervention for local recurrence while maintaining comparable rates of metastatic disease control relative to standard surgical approaches

Our study included patients with both adenocarcinoma and squamous cell carcinoma histologies, and we observed a substantial proportion of clinical complete responses across both groups. Notably, among patients with adenocarcinoma managed without surgery, survival outcomes were comparable to those of patients with squamous cell carcinoma undergoing active surveillance. These findings support the growing evidence that active surveillance may be a viable treatment strategy not only for squamous cell carcinoma—as is often assumed—but also for adenocarcinoma. It is important to note that current clinical practice for esophageal adenocarcinoma has shifted away from neoadjuvant chemoradiotherapy. Following the ESOPEC trial, perioperative chemotherapy with the FLOT regimen has emerged as the new standard of care, demonstrating superior overall and progression-free survival [16]. Consequently, this patient population is no longer the primary target for chemoradiotherapeutic strategies, as had been the case in the past. In the recently published SANO trial, two-thirds of patients with a cCR were found to have adenocarcinoma, with survival outcomes equivalent to those observed in the surgery arm. Our results therefore further support the consideration of active surveillance as a valid option in selected patients with adenocarcinoma.

Although our study is retrospective in nature, its findings are reinforced by a recently published prospective study, the SANO trial, that investigated the same clinical question and reached similar conclusions [23]. This concordance strengthens the overall validity of the observed association and highlights the robustness of the evidence. Furthermore, by including a broader, real-world cohort, the retrospective design enhances external validity and provides context to findings from randomized controlled trials.

This study has several important limitations. First, its retrospective, single-center, cohort study design introduces inherent methodological weaknesses, including selection bias, information bias, and limited internal validity. Known or unknown confounders may have influenced outcomes. Specifically, the study is subject to confounding by indication: patients managed with surveillance were not randomly assigned but were more likely to have baseline characteristics—such as comorbidities or frailty—that influenced both treatment allocation and outcomes. This inherent imbalance complicates causal interpretation of the observed outcomes, as patients who were older or had poorer baseline fitness were disproportionately assigned to the surveillance group, potentially influencing the results independently of the treatment strategy itself. Additionally, the relatively small sample size and limited follow-up duration reduce the statistical power to detect subtle but clinically meaningful differences and may also lead to unstable effect estimates, potentially exaggerating the true differences between treatment strategies. Finally, while this study adds to the growing body of evidence supporting active surveillance in selected patients, definitive conclusions require validation through prospective randomized controlled trials. The ongoing NEED trial is a phase III RCT assessing whether dCRT with salvage esophagectomy is non-inferior to nCRT with mandatory surgery for OS, and superior in HRQoL.

## 5. Conclusions

Surgery provides a survival benefit in younger patients with esophageal cancer who achieve clinical complete response after chemoradiotherapy, whereas surveillance is a safe and appropriate option for older patients, underscoring the need for age-adapted treatment strategies. Future prospective randomized studies are needed to validate age-adapted treatment strategies and to better define which patients may safely benefit from active surveillance. In addition, future research should incorporate standardized restaging criteria and advanced molecular or imaging biomarkers to more accurately identify true complete responders.

## Figures and Tables

**Figure 1 cancers-17-03926-f001:**
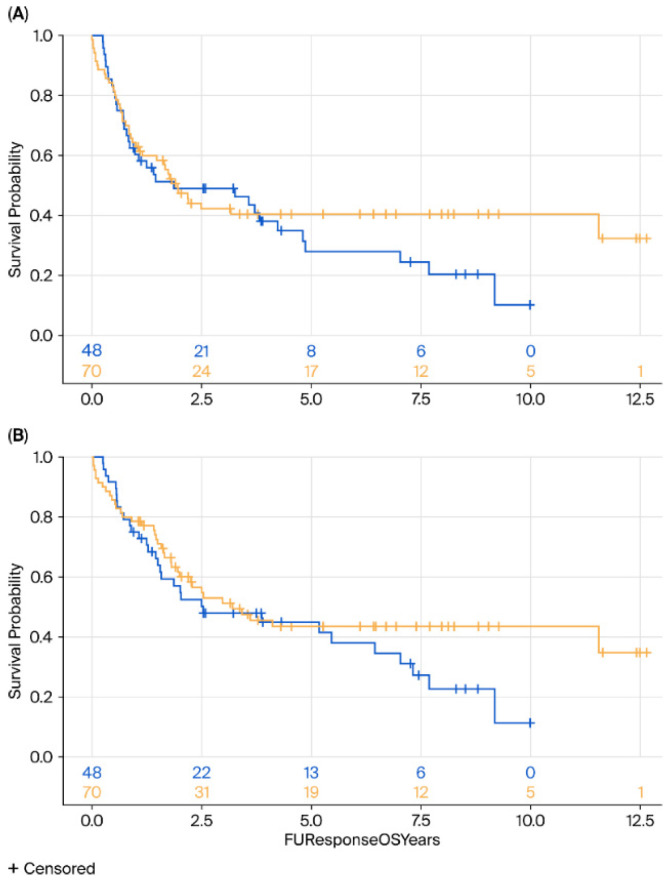
Kaplan–Meier survival analysis of DFS (**A**) and OS (**B**) of surgery (yellow) vs. surveillance (blue) groups. DFS, disease-free survival; OS, overall survival.

**Figure 2 cancers-17-03926-f002:**
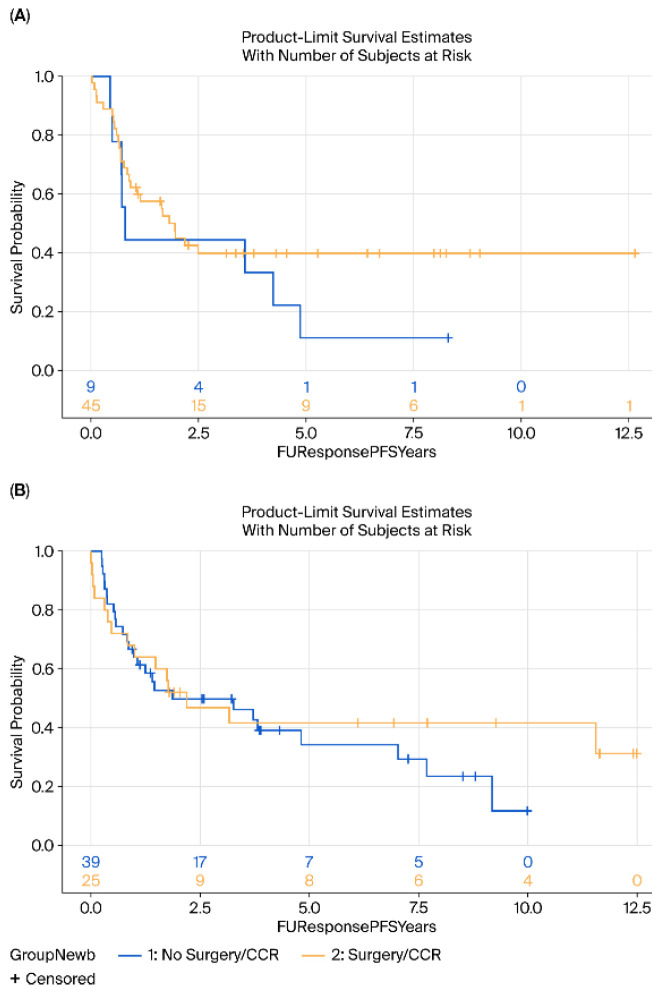
Kaplan–Meier survival analysis of DFS (**A**) Adenocarcinoma and (**B**) Squamous cell carcinoma of surgery (yellow) vs. surveillance (blue) groups. DFS, disease-free survival.

**Figure 3 cancers-17-03926-f003:**
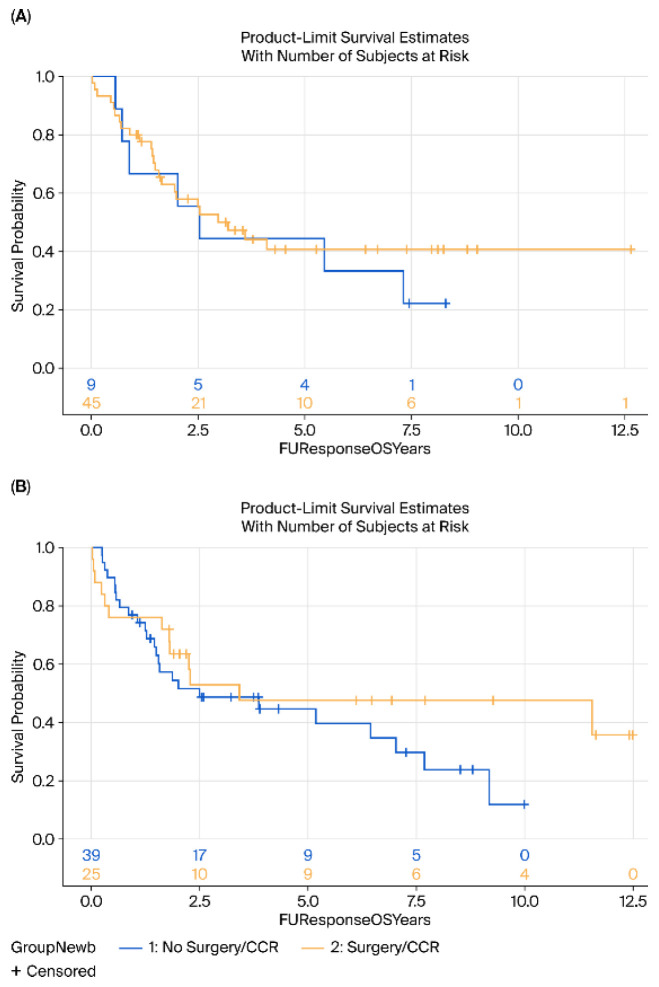
Kaplan–Meier survival analysis of OS (**A**) Adenocarcinoma and (**B**) Squamous cell carcinoma of surgery (yellow) vs. surveillance (blue) groups. OS, Overall-survival.

**Figure 4 cancers-17-03926-f004:**
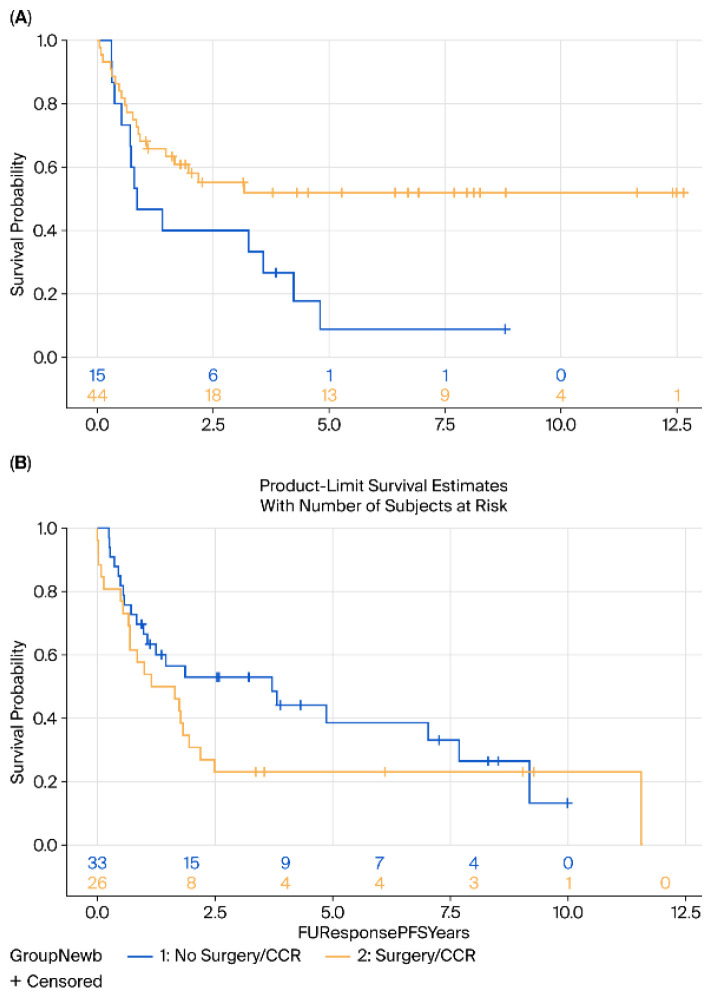
Kaplan–Meier survival analysis of (**A**) DFS and (**B**) OS of patients aged 70 and younger comparing surgery (yellow) vs. surveillance (blue).

**Figure 5 cancers-17-03926-f005:**
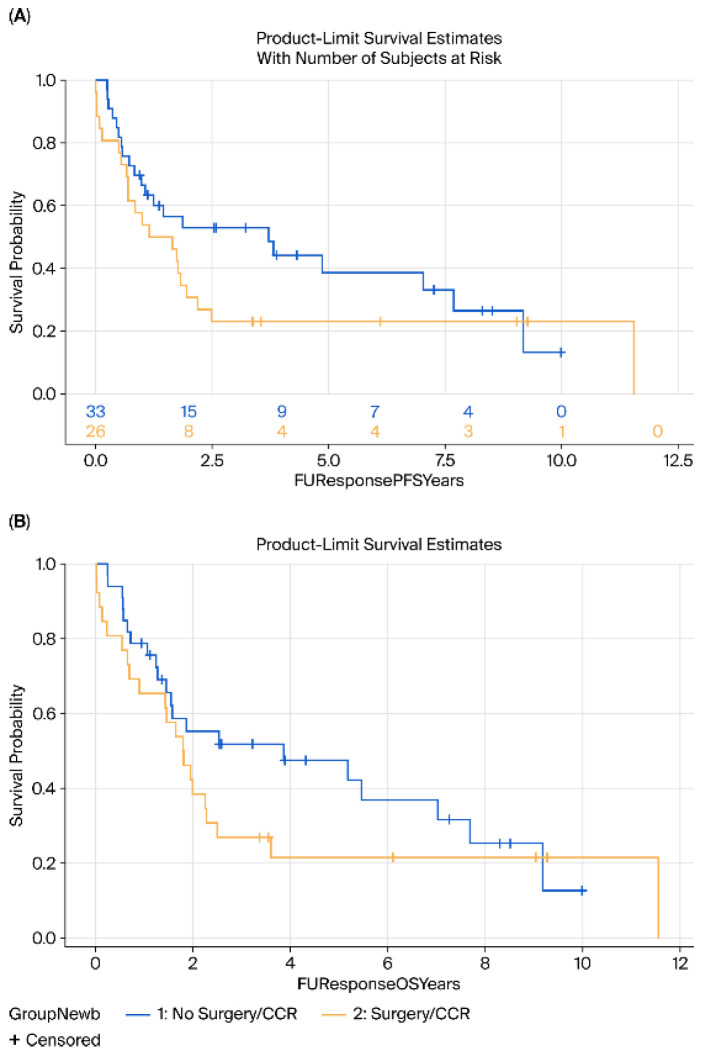
Kaplan–Meier survival analysis of (**A**) DFS and (**B**) OS of patients above 70 years comparing surgery (yellow) vs. surveillance (blue).

**Table 1 cancers-17-03926-t001:** Surgery versus surveillance.

Variable	Active Surveillance (*N* = 48)	Surgery (*N* = 70)	Overall (*N* = 118)	*p*-Value
Age (y, SD)	74.4 ± 9.78	65.38 ± 10.27	79.05 ± 10.97	<0.0001
Age ≥ 70 (%)	33 (68%)	26 (37%)	59 (50%)	
Gender (%Male) Female	21 (43%)	42 (60%)	63 (53%)	0.09
Histology (%SCC) ADENOCARCINOMA	81.2	35.7	54.24	<0.001
Tumor location (%)				0.02
Upper	6.25	1.43	3.39	
Mid	37.5	18.5	26.2	
Distal	39.5	47.1	44.07	
GEJ	16.6	32.8	26.2	
T3–4 (%)	75	90	83.9	0.04
N0 (%)	31.2	30	30.5	1
N1 (%)	66.6	68.5	67.8	
CROSS protocol (%)	47.9	42.8	44.9	<0.001
CISPLATIN + 5FU (%)	41.6	14.2	25.4	
Dose 41.4 gy (%)	31.2	41.4	38.1	0.44
Dose 50.4 gy (%)	64.5	58.5	61.5	0.567
Recurrence (%)	37.5	37.1	37.3	1
Distant (%)	25	32.8	28.9	0.47
Local (%)	12.5	2.8	7.65	0.09

Baseline clinical and treatment characteristics of patients managed with active surveillance versus esophagectomy. Continuous variables are presented as mean ± SD, and categorical variables as percentages. Values represent comparisons between treatment groups. (Abbreviations: SCC, squamous cell carcinoma; GEJ, gastroesophageal junction; T3–4, tumor stage T3–4; N0/N1, nodal stage 0/1; CROSS, chemoradiotherapy per the CROSS regimen; Cisplatin + 5FU, cisplatin plus 5-fluorouracil; Gy, Gray; Recurrence, any recurrence; Distant, distant recurrence; Local, locoregional recurrence).

## Data Availability

The data underlying this study consist of retrospective, patient-level clinical information and cannot be shared publicly due to institutional and ethical restrictions.

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
