# Peer review of "Esophagectomy Versus Active Surveillance After Complete Response in Locally Advanced Esophageal Cancer: Retrospective Analysis"

_cancers, 2025, doi:10.3390/cancers17243926_

Round 1

Reviewer 1 Report

Comments and Suggestions for Authors

The authors of the study compared survival outcomes in esophageal cancer patients with clinical complete response (cCR) following neoadjuvant chemoradiotherapy (nCRT) who underwent either surgery or surveillance. The authors found that younger patients (<70 yo) had significant survival benefits with surgery, whereas in older patients surveillance was safe and associated with similar survival outcomes, as compared to surgery.

Congratulations to the authors for a well written manuscript on a timely topic. Limitations of the study are adequately addressed. I have the following comments:

  • A key finding of your study was that 53% of the surgical patients had residual pathological disease! Clearly, current staging modalities have significant limitations for determining complete response. Although this issue is shortly discussed in the manuscript, further evidence on sensibility and specificity of diagnostic modalities for assessing response should be added. What about ctDNA? Different centers have included ctDNA in their surveillance protocol and this should also be further discussed.
  • Although I understand that is not part of the scope of the manuscript, information regarding surgical morbidity on your surgical patients might be useful for the readership. Any significant difference on morbidity between younger and older patients? This information is also helpful during the decision-making process between surgery or surveillance.

Author Response

Comment 1:

A key finding of your study was that 53% of the surgical patients had residual pathological disease! Clearly, current staging modalities have significant limitations for determining complete response. Although this issue is shortly discussed in the manuscript, further evidence on sensibility and specificity of diagnostic modalities for assessing response should be added. What about ctDNA? Different centers have included ctDNA in their surveillance protocol and this should also be further discussed.

We completely agree; this is a highly relevant and timely issue. We have incorporated a dedicated statement addressing this point in the revised paragraph:

Among patients in the surgery group, a pathological complete response (pCR) was observed in 47%, indicating that current restaging modalities—such as CT, PET-CT, and endoscopic biopsy—underestimated residual disease in nearly half of cases. This discordance between cCR and pCR has been widely reported, with literature citing pCR rates among cCR patients ranging from 24% to 73%, likely due to heterogeneity in restaging approaches [12,17-18]. These findings highlight the need for standardized, rigorous restaging algorithms to more reliably identify true responders and guide treatment decisions. Recent evidence further suggests that even among patients achieving pCR, minimal residual disease may still be present when evaluated with advanced molecular or imaging-based techniques. A recent study demonstrated that circulating tumor DNA (ctDNA)–based monitoring can detect molecular residual disease in patients classified as pCR, identifying individuals at higher risk for recurrence despite histopathologic clearance. Such tools may complement conventional restaging and ultimately improve selection of candidates for active surveillance[36].

Commet 2:

Although I understand that is not part of the scope of the manuscript, information regarding surgical morbidity on your surgical patients might be useful for the readership. Any significant difference on morbidity between younger and older patients? This information is also helpful during the decision-making process between surgery or surveillance.

We have added data on postoperative surgical complications for the study population to the manuscript; please find the details attached below:

Among the 66 patients who underwent esophagectomy, postoperative complications were observed in several domains. Pulmonary complications occurred in 10 patients (15.1%), with pneumonia accounting for 15% of all reported complications. Respiratory failure was documented in 2 patients (3.0%). Anastomotic leakage was identified in 20% of surgical patients (13 of 66 patients). In addition, deep vein thrombosis (DVT) occurred in 3 patients (4.5%). These findings reflect the substantial morbidity associated with esophagectomy in this cohort.

Reviewer 2 Report

Comments and Suggestions for Authors

Dear Author, thank you for sharing your research.
In this retrospective study from the Davidoff Cancer Center (2013–2023) involving 252 patients with locally advanced esophageal cancer treated with neoadjuvant chemoradiotherapy (nCRT), 118 achieved a clinical complete response (cCR). Seventy underwent surgery and 48 were managed with surveillance. Five-year overall survival was similar between groups (49% vs 48%). In patients aged ≤70 years, surgery provided a significant survival benefit, whereas in those >70 years, surveillance achieved comparable or better outcomes. Older age and female sex were independent predictors of poorer prognosis.
I found the manuscript interesting both for its conclusions and for the sample size. The results should be confirmed by further studies, but they represent a cautious and at the same time useful approach to disease management. I have no additional modifications to suggest.
Kind regards.

Author Response

I found the manuscript interesting both for its conclusions and for the sample size. The results should be confirmed by further studies, but they represent a cautious and at the same time useful approach to disease management. I have no additional modifications to suggest.
Kind regards.

No further comments

Thank you

Reviewer 3 Report

Comments and Suggestions for Authors

Dear authors, in order to improve the quality of the paper, I suggest the following corrections.

The title is unclear to the reader, so I suggest an extension to make it informative.
In the structured summary, you state EC, but you do not state the histological type. I suggest a correction, stating the histology of esophageal carcinoma. 
Line 48...please delete numbers 3-10. Expand the number of keywords with MesH terms. 
Please state in the Introduction which histological type of esophageal carcinoma you are referring to, squamous cell or adenocarcinoma, especially when you state the epidemiology of line 52.
Lines 53-56, please list current guidelines, not just individual publications. I recommend citing national and European, and you can also list some other global guidelines. 
When citing references 8-10, please expand them with more recent references.
Line 78. Please move the sentence to the end of the discussion.

Section 2. Please list the histology of the cancer. Also, write more clearly the research interval, including the month, and not just the year. 
The inclusion criteria should be written more clearly and precisely.
Please provide the Ethics Committee protocol number, as it is required for this type of study.
For endoscopy, please indicate the manufacturer of the equipment, the experience of the gastroenterologists who performed the upper endoscopy, and how many endoscopists were involved in the study. 
In section 2, materials and methods, please state the criteria used to divide patients into groups depending on whether they will be treated surgically or with observation only. Do you have informed consent from the patient?
In part 2.1. you did not state the manufacturer of the statistical analysis program. You also did not state which value is considered statistically significant. 
Table 1 does not have a Legend at the bottom, and you should state the abbreviations. 
Only in the results do you mention the histological types of esophageal carcinoma for the first time. I advise you to state the etiological, epidemiological and therapeutic differences of these two types of esophageal carcinoma in the introduction.
Expanded list of abbreviations. 
Please also include other parameters of the subjects in the results, such as comorbidities, smoking status and other parameters.
In the final part, you did not list author contributions, Funding and all other necessary parts according to the instructions for authors,
Also, the reference list is not written in accordance with the instructions. 10 authors and collaborators are listed.
Please correct this.
With the aim of improving the visibility of the article, it would be a suggestion to make an illustration for the graphic abstract.

Kind Regards

Author Response

The title is unclear to the reader, so I suggest an extension to make it informative
In the structured summary, you state EC, but you do not state the histological type. I suggest a correction, stating the histology of esophageal carcinoma 
Line 48...please delete numbers 3-10. Expand the number of keywords with MesH terms. 
Please state in the Introduction which histological type of esophageal carcinoma you are referring to, squamous cell or adenocarcinoma, especially when you state the epidemiology of line 52.
Lines 53-56, please list current guidelines, not just individual publications. I recommend citing national and European, and you can also list some other global guidelines. 
When citing references 8-10, please expand them with more recent references.
Line 78. Please move the sentence to the end of the discussion.

Section 2. Please list the histology of the cancer. Also, write more clearly the research interval, including the month, and not just the year. 
The inclusion criteria should be written more clearly and precisely.
Please provide the Ethics Committee protocol number, as it is required for this type of study.

Dear Editor,
I have addressed all the comments you noted above and have made the requested revisions to the manuscript accordingly

For endoscopy, please indicate the manufacturer of the equipment, the experience of the gastroenterologists who performed the upper endoscopy, and how many endoscopists were involved in the study. 

This study was conducted at a high-volume tertiary medical center specializing in complex surgical procedures, including esophagectomy. The gastroenterologists involved in patient evaluation and follow-up are among the most experienced in the country, recognized national leaders in advanced endoscopy, and utilize state-of-the-art endoscopic technologies. Approximately three expert gastroenterologists with dedicated expertise in this field were involved in the management of these patients.

In section 2, materials and methods, please state the criteria used to divide patients into groups depending on whether they will be treated surgically or with observation only. Do you have informed consent from the patient?

Because this was a retrospective study, informed consent for research was not required. Similarly, group allocation was not predetermined; as described in the manuscript, patients who were surgical candidates proceeded to esophagectomy, whereas those who declined surgery or were deemed inoperable continued with active surveillance. This design reflects the non-randomized nature of the study, which inherently introduces selection bias. We acknowledge this limitation and discuss its implications in the manuscript’s Discussion and Limitations sections.

In part 2.1. you did not state the manufacturer of the statistical analysis program. You also did not state which value is considered statistically significant. 

All statistical tests were two-sided, and a p-value <0.05 was considered statistically significant. Analyses were performed using SAS software, version 4.9 (SAS Institute Inc., Cary, NC, USA).

Table 1 does not have a Legend at the bottom, and you should state the abbreviations. 
Only in the results do you mention the histological types of esophageal carcinoma for the first time. I advise you to state the etiological, epidemiological and therapeutic differences of these two types of esophageal carcinoma in the introduction.

Expanded list of abbreviations.

Please also include other parameters of the subjects in the results, such as comorbidities, smoking status and other parameters. 

I have addressed all the comments you noted above and have made the requested revisions to the manuscript accordingly

In the final part, you did not list author contributions, Funding and all other necessary parts according to the instructions for authors, 
Also, the reference list is not written in accordance with the instructions. 10 authors and collaborators are listed.Please correct this. 

I have attached a revised reference list along with the full list of authors for the manuscript, prepared according to your requirements.

With the aim of improving the visibility of the article, it would be a suggestion to make an illustration for the graphic abstract.

We have submitted the manuscript for editorial processing and are currently awaiting the finalized graphical abstract.

Thank you

Reviewer 4 Report

Comments and Suggestions for Authors

Comment 1:
The study provides interesting findings, but the retrospective design introduces clear selection bias. The authors should acknowledge that older or less fit patients were more likely to be placed in the surveillance group, which could affect outcomes.

Comment 2:
The paper focuses only on survival, but omits postoperative complications or quality-of-life data. Including these aspects would strengthen the conclusion that surveillance is safer for elderly patients.

Comments on the Quality of English Language

Comment 1:
The study provides interesting findings, but the retrospective design introduces clear selection bias. The authors should acknowledge that older or less fit patients were more likely to be placed in the surveillance group, which could affect outcomes.

Comment 2:
The paper focuses only on survival, but omits postoperative complications or quality-of-life data. Including these aspects would strengthen the conclusion that surveillance is safer for elderly patients.

Author Response

Comment 1:The study provides interesting findings, but the retrospective design introduces clear selection bias. The authors should acknowledge that older or less fit patients were more likely to be placed in the surveillance group, which could affect outcomes.

Agreed. We are fully aware of the inherent bias in this study, and this is appropriately addressed in the limitations section. However, it is important to note that despite the older age and overall “less favorable” clinical profile of patients in the active surveillance group, their survival outcomes were not compromised and were comparable to those who underwent surgery. These points are clearly discussed in both the Discussion and the Limitations sections of the manuscript.

This study has several important limitations. First, its retrospective, single-center, cohort study design introduces inherent methodological weaknesses, including selection bias, information bias, and limited internal validity. Known or unknown confounders may have influenced outcomes. Specifically, the study is subject to confounding by indication: patients managed with surveillance were not randomly assigned but were more likely to have baseline characteristics—such as comorbidities or frailty—that influenced both treatment allocation and outcomes. This inherent imbalance complicates causal interpretation of the observed outcomes, as patients who were older or had poorer baseline fitness were disproportionately assigned to the surveillance group, potentially influencing the results independently of the treatment strategy itself.

Comment 2:
The paper focuses only on survival, but omits postoperative complications or quality-of-life data. Including these aspects would strengthen the conclusion that surveillance is safer for elderly patients.

Agreed. We have now added data regarding postoperative complications to the manuscript. Quality of life was not an outcome assessed in our study, and therefore we are unable to provide additional information on this measure.

Among the 66 patients who underwent esophagectomy, postoperative complications were observed in several domains. Pulmonary complications occurred in 10 patients (15.1%), with pneumonia accounting for 15% of all reported complications. Respiratory failure was documented in 2 patients (3.0%). Anastomotic leakage was identified in 20% of surgical patients (13 of 66 patients). In addition, deep vein thrombosis (DVT) occurred in 3 patients (4.5%). These findings reflect the substantial morbidity associated with esophagectomy in this cohort.

Round 2

Reviewer 3 Report

Comments and Suggestions for Authors

Dear authors

The quality of the manuscript increased significantly after the corrections were made.
Please put retrospective analysis in the title of the paper.
Also, please write 2 sentences related to future directions in the conclusion.
Please attach a graphic absctract.

Best regards